# Breaking Determinism: Fuzzy Modeling of Sequential Recommendation Using Discrete State Space Diffusion Model

**Wenjia Xie  Hao Wang**[§][*] **Luankang Zhang  Rui Zhou  Defu Lian  Enhong Chen**

University of Science and Technology of China & State Key Laboratory of Cognitive Intelligence
`xiaohulu@mail.ustc.edu.cn,wanghao3@ustc.edu.cn,zhanglk5@mail.ustc.edu.cn,`
`zhou_rui@mail.ustc.edu.cn,liandefu@ustc.edu.cn,cheneh@ustc.edu.cn`

## Abstract

Sequential recommendation (SR) aims to predict items that users may be interested in based on their historical behavior sequences. We revisit SR from a novel information-theoretic perspective and find that conventional sequential modeling methods fail to adequately capture the randomness and unpredictability of user behavior. Inspired by fuzzy information processing theory, this paper introduces the DDSR model, which uses fuzzy sets of interaction sequences to overcome the limitations and better capture the evolution of users' real interests. Formally based on diffusion transition processes in discrete state spaces, which is unlike common diffusion models such as DDPM that operate in continuous domains. It is better suited for discrete data, using structured transitions instead of arbitrary noise introduction to avoid information loss. Additionally, to address the inefficiency of matrix transformations due to the vast discrete space, we use semantic labels derived from quantization or RQ-VAE to replace item IDs, enhancing efficiency and improving cold start issues. Testing on three public benchmark datasets shows that DDSR outperforms existing state-of-the-art methods in various settings, demonstrating its potential and effectiveness in handling SR tasks.

## 1 Introduction

For a long time, sequential recommendation (SR) has been attracting increasing attention due to its excellent performance and significant commercial value (Chen et al. [2020], Qiu et al. [2021], Yin et al. [2024b], Han et al. [2024]). Unlike traditional collaborative filtering or certain graph-based methods (Wang et al. [2019], Zhang et al. [2024], Wang et al., Tong et al. [2024]), SR systems emphasize the inherent dynamic behaviors of users rather than relying solely on structured data (Chen et al. [2022], Ma et al. [2020], Cen et al. [2020]). This approach enhances the accuracy of personalized recommendations, allowing for more precise tracking of changes in user interests and needs. Typical deep learning-based SR models, such as those utilizing CNN, RNN, and Transformer architectures (Tang and Wang [2018], Hidasi et al. [2015], Kang and McAuley [2018b]), have achieved remarkable success in modeling user historical interaction data.

However, these methods are formalized models based on a narrow information theory assumption (Shannon [1948]), which only acknowledges determinism (Rosas et al. [2020]). They assume that all phenomena strictly adhere to mechanical laws and that the states of motion of objects at different times can be uniquely determined. In reality, however, user behavior is characterized by randomness and unpredictability. They might change their mind about buying a down jacket due to a sudden warm-up, or they might impulsively buy desserts due to a breakup. As illustrated on the left in Figure 1, a user's interest at any given moment might be focused on 'some items' with blurred boundaries, only converging finally when the user makes a selection.

---

[*]Hao Wang is the corresponding author.

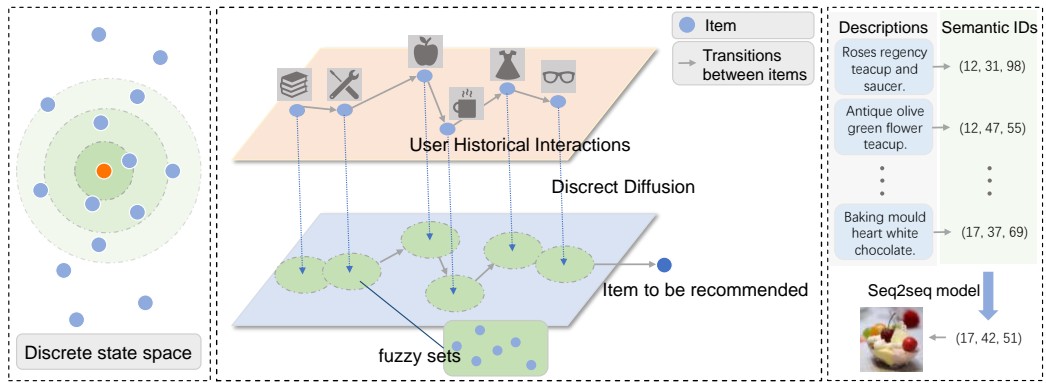

**Figure 1:** Illustration of DDSR constructing fuzzy sets and incorporating semantic IDs to enhance sequential recommendations. In real-world scenarios, a user's final choice often reflects their immediate interests (left subfigure). We reconstruct the true evolution of interests by constructing fuzzy sets for each item in the interaction sequence (middle subfigure). The right subfigure provides an overview of the process of generating semantic IDs for recommendations based on item-related descriptions.

Although increasing the sample size is an effective strategy to address the above issue, in reality, the data in recommendation systems is usually quite sparse (He and McAuley [2016]), limiting the practicality of this strategy. Inspired by the theory of fuzzy information processing (Tanaka et al. [1976], Tanaka and Sommer [1977]), we believe that making the absolute membership relations in traditional sets more flexible is another effective way to solve the problem. In other words, it is not necessary to strictly limit the modeling of user interests to the items they have interacted with. Therefore, we propose using a diffusion model (Ho et al. [2020]) for fuzzy modeling of user interests, which enhances the model's performance by introducing perturbations during the training process.

We have noticed existing work that introduces diffusion models into SR (Xie et al. [2024]), such as DiffuRec (Li et al. [2023b]) and DreamRec (Yang et al. [2023]), which focus on Gaussian diffusion processes operating within a continuous state space. They add Gaussian noise to the embedded representations of candidate items for recommendation through a forward diffusion process until the noise reaches a pure state (standard normal distribution). Subsequently, they iteratively sample from this noise using a reverse denoising process guided by historical interaction information to recover meaningful representations and recommend items most similar to these representations.

However, unlike our desire to fuzzily model interaction sequences, the aforementioned methods follow the form of diffusion models in the image domain and operate on candidate items. They introduce the crucial sequence information merely as conditional information, without leveraging the diffusion model's performance on it. On the other hand, these methods relax discrete interaction data into a continuous space and introduce noise, which may lead to distortion or loss of meaning in the original discrete space, as the addition of noise could push data points away from any meaningful discrete state. Therefore, we hope that state transitions occur under discrete conditions for the entire interaction sequence, which is discrete diffusion. Based on this, we have proposed our DDSR (Discrect Diffusion Sequential Recommendation model), which uses a directed graph to model sequential recommendation. In this model, all interaction items are viewed as nodes, and transitions between items are treated as directed edges. Discrete diffusion is used to enable structured transitions of nodes, with the resulting new sets treated as fuzzy sets, as shown in the middle of Figure 1. By designing the transition matrix, we can achieve uniform transitions or importance-based transitions for the nodes, ensuring controllability. In Section 3.3, we theoretically demonstrate the reliability and effectiveness of modeling on these generated fuzzy sets, based on the principles of information diffusion. During the inference stage, we refer to the sampling formula for discrete diffusion but start from the historical interaction sequence rather than from noise, iteratively generating refined results.

Furthermore, we have found that the excessive number of items involved in the recommendation problem leads to a high-dimensional transition matrix, resulting in inefficient diffusion transitions. Additionally, item IDs themselves do not contain any prior information, which poses a challenge in determining beneficial transition directions. To address this issue, we have further introduced semantic tags to replace meaningless item IDs, using quantization techniques and VQ-VAE to derive

these tags from semantic information, thus reducing the size of the discrete space. We will provide specific details on how this can be achieved in 4.1, and a vivid illustration of this is given on the right side of Figure 1. Simultaneously, the introduction of semantic information has enhanced the model's generalization capability and effectively solved the cold start problem. We conducted extensive experiments on three public benchmark datasets, comparing DDSR with several state-of-the-art methods. The results demonstrate that DDSR significantly outperforms baseline methods in various settings and effectively handles cold-start recommendations.

## 2 Related Work

### 2.1 Sequence Recommendation

SR suggests potential subsequent items based on users' historical interaction records (Yin et al. [2023], Wang et al. [2024]). Early research primarily relied on Markov chains and matrix factorization techniques for recommendation (He et al. [2016]). However, with the development of deep learning, efforts such as GRU4Rec (Hidasi et al. [2015]), Caser (Tang and Wang [2018]), and others have focused on designing neural network models to capture sequential dependencies in user behavior sequences. The introduction of the Transformer architecture (Vaswani et al. [2017]) in SASRec (Kang and McAuley [2018b]) pioneered SR and quickly became the mainstream method in the field. Additionally, BERT4Rec (Sun et al. [2019]) utilizes bidirectional encoders to capture bidirectional dependencies in sequences, using a masked language model to predict the user's next action.

Recent studies have shown that high-quality high-dimensional embeddings are crucial for obtaining accurate recommendation results (Hou et al. [2022], Wang et al. [2021]). To this end, researchers are striving to leverage the rich attribute information of items to improve data representation. For example, TransFM (Pasricha and McAuley [2018]) introduces arbitrary real-valued features through factorization machines, while S3-Rec (Zhou et al. [2020]) designs four self-supervised learning tasks as pre-training objectives to learn context-aware data representations with attribute awareness. Furthermore, researchers like Hou et al. [2022], Zhao [2022], Harte et al. [2023] further utilize pre-trained language models (Yin et al. [2024a]) to process item description texts, obtaining universal item representations with rich semantic information to enhance the performance (Wu et al. [2024]). VQ-Rec (Hou et al. [2023]) and TIGER (Rajput et al. [2024]) further employ quantization techniques (Jacob et al. [2018]) and RQ-VAE (Lee et al. [2022]) to obtain tokenized semantic IDs for recommendations, replacing semantic embeddings.

### 2.2 Discrete Diffusion Models

Diffusion models, inspired by non-equilibrium thermodynamics, have been introduced and demonstrated significant results in fields such as computer vision, sequence modeling, and audio processing (Dhariwal and Nichol [2021], Rasul et al. [2021], Ho et al. [2022]). Most diffusion models are based on the Denoising Diffusion Probabilistic Model (DDPM) proposed by Ho et al. [2020], as well as the score-based generative models (SGMs) proposed by Song et al. [2020], targeting continuous data domains. We provide detailed descriptions of DDPM and SGMa in Appendix D to facilitate comparisons with the discrete diffusion approach we employ. Diffusion models in discrete state space are first described in Sohl-Dickstein et al. [2015] and later applied to text and image domains in D3PMs (Austin et al. [2021]). VQ-Diffusion (Gu et al. [2022]) utilizes them to eliminate unidirectional bias in text-to-image generation.

## 3 Discrete Diffusion Process of DDSR

In this section, we present the problem definition (Section 3.1) and illustrate how item sequences undergo discrete diffusion to obtain the corresponding fuzzy sets (Section 3.2). Finally, the effectiveness of this fuzzy modeling is theoretically demonstrated (Section 3.3). Please note that the actual diffusion and inference in DDSR occur at the semantic ID level, but this chapter discusses items.

### 3.1 Problem Statement

Let $\mathcal{U}$ be the set of users and $\mathcal{V}$ be the set of discrete items in the dataset, $|\mathcal{U}|$ and $|\mathcal{V}|$ represent the number of elements in their respective sets. For each user $\mathbf{u} \in \mathcal{U}$, $v_{1:n-1} = [v_1, v_2, \ldots, v_{n-1}]$

represents his historical interaction sequence sorted by timestamp. The goal of the model is to predict the next item $v_n$ that the user is most likely to interact with. To facilitate better discrete diffusion and for the convenience of subsequent theoretical derivations, we model each user's interaction sequence as a directed graph $\mathcal{G}^u$. In this graph, each item represented by a semantic ID is regarded as a node, while transitions between items are viewed as directed edges. Specifically, an edge exists from $v_i$ to $v_j$ if and only if $v_j$ is the next item interacted with by the user after $v_i$.

## 3.2 Node Diffusion Transition

A typical diffusion model transforms data $x_0 \sim q(x_0)$ into a sequence of gradually noisier latent variables $x_{1:T} = x_1, x_2, ..., x_T$ via forward process $q(\boldsymbol{x}_{1:T}|\boldsymbol{x}_0) = \prod_{t=1}^T q(\boldsymbol{x}_t|\boldsymbol{x}_{t-1})$. In diffusion models within continuous state spaces, the forward distribution is typically set with $q(\boldsymbol{x}_t|\boldsymbol{x}_{t-1}) = \mathcal{N}\left(\boldsymbol{x}_t|\sqrt{1-\beta_t}\boldsymbol{x}_{t-1}, \beta_t\boldsymbol{I}\right)$ as a hyperparameter controlling the level of noise added at each step. As the number of time steps $T$ approaches infinity, $x_T$ converges to a standard Gaussian distribution. Beyond the limitations mentioned above, information loss due to diffusion into pure noise is another reason for unstable training and inadequate alignment of continuous diffusion with SR.

Continuous diffusion always operates on embeddings, while in diffusion models within discrete state spaces, categories are directly transformed. Transition matrices $[\boldsymbol{Q}_t]_{ij} = q(x_t = j|x_{t-1} = i)$ are used to describe the probability of single-step diffusion transitions, where $i$ and $j$ represent categories within the domain. Denoting the one-hot version of $x$ with the row vector $\boldsymbol{x}$ (bold), then the one-step transition probabilities can be expressed as:

$$q(\boldsymbol{x}_t|\boldsymbol{x}_{t-1}) = \text{Cat}(\boldsymbol{x}_t; \boldsymbol{p} = \boldsymbol{x}_{t-1}\boldsymbol{Q}_t), \tag{1}$$

where $Cat(\boldsymbol{x}; p)$ is the categorical distribution corresponding to the one-hot row vector $\boldsymbol{x}$ with probabilities given by the row vector $p$, and $\boldsymbol{x}_{t-1}\boldsymbol{Q}_t$ is understood as a row vector-matrix product. Starting from $\boldsymbol{x}_0$, we obtain the following $t$-step marginal and posterior at time $t-1$:

$$q(\boldsymbol{x}_t|\boldsymbol{x}_0) = \text{Cat}\left(\boldsymbol{x}_t; \boldsymbol{p} = \boldsymbol{x}_0\overline{\boldsymbol{Q}}_t\right), \quad \text{with} \quad \overline{\boldsymbol{Q}}_t = \boldsymbol{Q}_1\boldsymbol{Q}_2\dots\boldsymbol{Q}_t, \tag{2}$$

We take the set $\mathcal{V}$ as the domain in SR. Each $v_i$ in the interaction sequence is represented as a one-hot encoding $\boldsymbol{x}_i^0$. Using the transition form defined by 2 enables transitions to other nodes, denoted as $\boldsymbol{x}_i^t$, in the domain with a certain probability at any time step $t$. Unlike continuous diffusion, which only allows noise addition, discrete diffusion models offer the advantage of controlling the data blurring process by selecting the transition matrix. Here, we present two strategies to select transition matrices, i.e. Uniform transition and Importance transition.

**Uniform transition.** Similar to the study by Hoogeboom et al. [2021], the natural idea is to maintain nodes with a certain probability $\beta_t \in (0, 1)$ unchanged. In contrast, in other cases, nodes are randomly transformed into any other node in the domain with equal probability $(1 - \beta_t)/(|\mathcal{V}| - 1)$. That is

$$[\boldsymbol{Q}_t]_{ij} = \begin{cases} (1-\beta_t)/(|\mathcal{V}|-1) & \text{if } i \neq j \\ \beta_t & \text{if } i = j \end{cases}. \tag{3}$$

Uniform transfer can be regarded as a special case of linear information allocation, thus theoretically affected by the size of the discrete space. It can compute the cumulative product $\overline{\boldsymbol{Q}}_t$ in closed form.

**Importance transition.** For data with certain prior knowledge, we propose transitioning between more similar nodes rather than uniformly transitioning to any other state, thus defining the matrix:

$$[\boldsymbol{Q}_t]_{ij} = \begin{cases} \dfrac{\exp\left(-d_{ij}^2/2\sigma^2\right)}{\sum_{v_k \in \mathbf{V}} \exp\left(-d_{ik}^2/2\sigma^2\right)} & \text{if } i \neq j \\ 1 - \sum_{k=0, k\neq i}^{|\mathbf{V}|-1}[\boldsymbol{Q}_t]_{ik} & \text{if } i = j \end{cases}. \tag{4}$$

Here, $d_{ij}$ represents the distance between item $v_i$ and $v_j$, calculated using the square of the Euclidean distance. The parameter $\sigma^2$ denotes the variance of the diffusion process. Consequently, the transition probabilities cannot be solved in closed form; instead, they can only be updated alongside the embeddings in the model. The importance transfer matrix adheres to the Gaussian information diffusion function $f(x) = \frac{1}{\sigma\sqrt{2\pi}}e^{-\frac{x^2}{2\sigma^2}}$. Therefore, it remains unaffected by the number of discrete points but necessitates the sample point distribution to closely resemble a Gaussian distribution.

This modeling approach appears to align more closely with our intuition, as a user's interests at a given moment often form a cluster of similar nodes (items). Only upon the user's final selection of an item does the 'neighborhood' converge to a single data point. This point represents the representative of the interest cluster and the ambiguity in information naturally dissipates. In recommendation tasks, we can only access the user's final choice at each moment, without knowledge of the interest cluster, reflecting incomplete knowledge. Regardless of the transition method employed, the key lies in transitioning the sample space from incomplete to complete, as detailed in the subsequent section.

### 3.3 Completeness and Reliability

Here, we aim to demonstrate that the discrete diffusion spaces generated by the two methods in Section 3.2 are completions of the original space, and the models constructed on these fuzzy sets are solvable and effective. We first provide the formal definition of a complete sample space.

**Definition 1.** *Let $W$ denote the sample space. For any sample $W \in \mathbf{W}$, if $W$ is complete, i.e., unbiased estimates can be obtained through certain mathematical processing, then $\mathbf{W}$ is called a complete sample space; otherwise, it is called an incomplete sample space.*

In SR, $W$ is a user's behavior sequence; $\mathbf{W}$ is all possible combinations of these behavior sequences; the domain $V$ is all items in the dataset. Datasets in SR systems do not form a complete sample space, as they often consist of incomplete interaction data and potential selection biases. The principle of information diffusion ensures that when the given sample is incomplete, there exist reasonable diffusion functions that can improve non-diffusion estimates. Below we define information diffusion.

**Definition 2.** *An information diffusion about a set $W$ is defined by a mapping $\mu : W \times V \to [0, 1]$, satisfying the following conditions:*

(1) $\forall w_j \in W$, if $v_j$ is the observed value of $w_j$, then $\mu(w_j, v_j) = \sup_{v \in V} \mu(w_j, v)$.
(2) $\forall w_j \in W$, $\mu(w_j, v)$ decreases as $\|v_j - v\|$ increases.
(3) $\forall w \in W$, $\sum_{\nu} \mu(w, v) \mathrm{d}v = 1$.

The diffusion estimates obtained using uniform transition and importance transition, as defined in Section 3.2, clearly adhere to Definition 2. To illustrate that the space resulting from discrete state transitions provides more information than the original state space, it is necessary to further demonstrate that this space serves as a completion of the original space. In other words, the new metric space is complete, with the original metric space serving as its dense subspace. This will be more precisely discussed in the following theorem.

**Theorem 3.1.** *After information diffusion, the subsequent space must be an entirely separable metric space. Any model constructed in this space will assuredly possess a solution.*

Proof in Appendix A. According to Theorem 3.1, since the space after information diffusion is equidistant isomorphism with the original space, it can be used to replace the sample space with insufficient information in SR to establish a model. On this complete space, predictive models are solvable, demonstrating that modeling on fuzzy sets is a reasonable and effective approach.

## 4 Learning and Inference of DDSR

### 4.1 Obtaining Semantic IDs

As mentioned in Section 3, the indices $i$ and $j$ in the transition matrix $[\mathbf{Q}_t]_{ij}$ represent categories in the discrete space, making $\mathbf{Q}_t$ a two-dimensional matrix with dimensions equal to the size of the discrete space. However, in recommendation tasks, the number of items involved can reach tens of thousands, posing a significant challenge in terms of computational resources if we were to use all item IDs as the discrete state space. Inspired by VQ-Rec and the recently proposed Tiger model by Google, we attempt to train recommendation models using semantic IDs instead of item IDs. A semantic ID is a codebook of length $m$. Assuming we set the size of the codebook to $K$, the entire codebook can represent $K^m$ categories. Though we set each code from a different codebook, the state space only needs $m * K$ nodes to store them. Additionally, the use of semantic IDs further introduces semantic information, addressing the scarcity of information inherent in recommendations, while also allowing the model to extend to unseen items, thus enabling cold-start recommendations. We provide the specific method for obtaining semantic IDs in the Appendix B.

---

**Algorithm 1** Training of DDSR.

---

**Input:** historical interaction sequence $v_{1:n-1} = c_{1:n-1;1:m}$; target item $v_n = c_{n;1:m}$; transition matrix $\boldsymbol{Q}_t$; Approximator $f_\theta(\cdot)$.
**Output:** well-trained Approximator $f_\theta(\cdot)$.
    While not converged do:
1: Sample Diffusion Time: $t \sim [0, 1, \ldots, T]$;
2: Calculate $t$-step transition probability: $\overline{\boldsymbol{Q}_t} = \boldsymbol{Q}_1 \boldsymbol{Q}_2 \cdots \boldsymbol{Q}_t$;
3: Convert $c_{n;1:m}$ to one-hot encoding $\boldsymbol{x}_{n;1:m}^0$;
4: Obtain the discrete state $x_{n;1:m}^t$ after $t$ steps by Equation 2, thereby obtaining the 'fuzzy set' $c_{1:n-1;1:m}^t$;
5: Modeling $c_{2:n;1:m}$ based on 'fuzzy sets' through Equation 5;
6: Take gradient descent step on $\nabla L_{CE} \left( \hat{c}_{2:n;1:m}, c_{2:n;1:m} \right)$.

---

 

---

**Algorithm 2** Inference of DDSR.

---

**Input:** historical sequence $c_{1:n-1;1:m}$; well-trained Approximator $f_\theta(\cdot)$; sampling step $T$.
**Output:** predicted target item $v_n$.
1: Let $\boldsymbol{x}_T = c_{1:n-1;1:m}$;
2: Let $t = T$;
3: **while** $t > 0$ **do**
4:     Use the trained $f_\theta(\cdot)$ to obtain predictions $\tilde{\boldsymbol{x}}_0$ with $\boldsymbol{x}_t$ and $t$ as inputs;
5:     Substitute $\tilde{x}_0$ into equation 7 to obtain the distribution of $t - 1$ step;
6: **end while**
7: $\tilde{v}_n = \boldsymbol{x}_0[-1; 1 : m]$;
8: if the same code project exists: $v_n = \tilde{v}_n$;
9: else: $v_n$ is the project in the space closest to $\tilde{v}_n$.

---

## 4.2 Model Training

After introducing the Semantic ID, we convert the historical interaction sequence $v_{1:n-1}$ into sequence $(c_{1,1}, \ldots, c_{1,m}; c_{2,1}, \ldots, c_{2,m}; \ldots; c_{n-1,1}, \ldots, c_{n-1,m})$, abbreviated as $c_{1:n-1;1:m}$. We convert them into one-hot encodings $(\boldsymbol{x}_{1,1}^0, \ldots, \boldsymbol{x}_{1,m}^0; \ldots; \boldsymbol{x}_{n-1,1}^0, \ldots, \boldsymbol{x}_{n-1,m}^0)$, which is considered as the initial state for discrete diffusion. Then we perform discrete diffusion through the state transition formula defined in 2 (for more details, see Section 3.2) to obtain the discrete state after $t$ steps $\boldsymbol{x}_{i,j}^t$, for any $i \in \{1, \ldots, n-1\}$ and $j \in \{1, \ldots, m\}$. Accordingly, the labels changes from $c_{i,j}$ to $c_{i,j}^t$. Then $(c_{1,1}^t, \ldots, c_{1,m}^t; \ldots; c_{n-1,1}^t, \ldots, c_{n-1,m}^t)$ forms a "fuzzy set" of $c_{1:n-1;1:m}$, denoted as $c_{1:n-1;1:m}^t$, which can also be viewed as the state $x_t$ of the diffusion transition at step $t$.

Considering the suitability of the Transformer for sequence-to-sequence tasks, along with its well-demonstrated effectiveness in modeling sequential dependencies, we use it with an embedding layer as Approximator $f_\theta(\cdot)$ to predict $c_{2:n;1:m}$ with $c_{1:n-1;1:m}^t$ as input. This approach differs from the common practice in diffusion models, which often focus on modeling noise. It aligns more closely with typical SR tasks that use $v_{1:n-1}$ to predict $v_{2:n}$, that is, the distribution $\tilde{p}_\theta(\tilde{\boldsymbol{x}}_0|\boldsymbol{x}_t)$. We have adopted sinusoidal time step embeddings, which are added after the embedding layer, allowing the model to capture information about the time steps. This process can be represented by:

$$\hat{c}_{2:n;1:m} = f_\theta(c_{1:n-1;1:m}^t, t). \tag{5}$$

Generally, the loss function of diffusion models is designed based on KL divergence, or it can be simplified to mean-squared error. However, guided by the theory of information diffusion, we choose to use a cross-entropy loss function, which is more suitable for recommendation tasks, to optimize our model without being constrained by the aforementioned methods.

**Table 1:** Detailed descriptions and statistics of datasets. 'Avg. length' represents the average length of item sequences, while 'Avg. num' indicates the average number of words in item text.

| Datasets | Users | Items | Interactions | Avg.length | Avg.num |
|---|---|---|---|---|---|
| Scientific | 8442 | 4385 | 59 427 | 7.04 | 182.87 |
| Office | 87 436 | 25 986 | 684 837 | 7.84 | 193.22 |
| Online Retail | 16 520 | 3469 | 519 906 | 26.90 | 27.80 |

### 4.3 Model Inference

In the inference phase, we aim to emulate the reverse process of the diffusion model, iteratively producing refined recommendation results. According to Bayes' theorem, we have

$$q(\boldsymbol{x}_{t-1}|\boldsymbol{x}_t, \boldsymbol{x}_0) = \frac{q(\boldsymbol{x}_t|\boldsymbol{x}_{t-1}, \boldsymbol{x}_0)q(\boldsymbol{x}_{t-1}|\boldsymbol{x}_0)}{q(\boldsymbol{x}_t|\boldsymbol{x}_0)} = \text{Cat}\left(\boldsymbol{x}_{t-1}; \boldsymbol{p} = \frac{\boldsymbol{x}_t\boldsymbol{Q}_t^\top \odot \boldsymbol{x}_0\overline{\boldsymbol{Q}}_{t-1}}{\boldsymbol{x}_0\overline{\boldsymbol{Q}}_t\boldsymbol{x}_t^\top}\right), \quad (6)$$

where $\odot$ represents the Hadamard product. Following the approach of Ho et al. [2020] and Hoogeboom et al. [2021], we employ the trained model $f_\theta(\cdot)$ as described in Section 4.2 to derive the distribution $\tilde{p}_\theta(\tilde{\boldsymbol{x}}_0|\boldsymbol{x}_t)$. Combining it with $q(\boldsymbol{x}_{t-1}|\boldsymbol{x}_t, \boldsymbol{x}_0)$, we obtain the following parameterized expression:

$$p_\theta(\boldsymbol{x}_{t-1}|\boldsymbol{x}_t) = \sum_{\tilde{\boldsymbol{x}}_0} q(\boldsymbol{x}_{t-1}, \boldsymbol{x}_t|\tilde{\boldsymbol{x}}_0)\tilde{p}_\theta(\tilde{\boldsymbol{x}}_0|\boldsymbol{x}_t). \quad (7)$$

For the historical interactions $v_{1:n-1}$, we use $c_{1:n-1;1:m}$ as $x_T$, and starting from $t = T$, we iteratively execute Equation 7 until $t = 0$. This parameterization also allows us to perform inference for $k$ steps at a time by predicting $p_\theta(\boldsymbol{x}_{t-k}|\boldsymbol{x}_t) = \sum q(\boldsymbol{x}_{t-k}, \boldsymbol{x}_t|\tilde{\boldsymbol{x}}_0)\tilde{p}_\theta(\tilde{\boldsymbol{x}}_0|\boldsymbol{x}_t)$, leading to efficiency improvements. After obtaining $\boldsymbol{x}_0$, we take its last component, which is a semantic ID of length m. If the corresponding item exists, we directly select that item; otherwise, we search for the item closest to it in the embedding space as the final recommendation result. The training and inference phase of DDSR are demonstrated in Algorithm 1 and Algorithm 2.

## 5 Experiment

### 5.1 Experiment Settings

**Datasets**. We employ three real-world datasets to evaluate the performance of our DDSR model. Following some works on text-based recommendation (Li et al. [2023a], Hou et al. [2022]), these datasets include two specific subcategories from the Amazon Reviews dataset (**Scientific** and **Office**), and a cross-platform dataset known as **Online Retail**, which operates from the UK. Following the method of Hou et al. [2022], we filter out users and items with fewer than five interactions. Subsequently, interaction behaviors within each sub-dataset are grouped by user and sorted chronologically. For the Amazon sub-datasets, product descriptions are formed by concatenating fields such as title, category, and brand, while for the Online Retail dataset, the description field is used. The product texts are truncated to 512 characters. Please refer to Table 1 for detailed descriptions of these datasets.

**Baselines**. We compare DDSR with eight state-of-the-art SR methods, including two conventional SR methods, three methods based on semantic information, and three generative SR methods:

**1). Conventional Baselines**: **SASRec** (Kang and McAuley [2018a]) utilizes a causal Transformer architecture with a self-attention mechanism to model user behavior. **BERT4REC** (Sun et al. [2019]) proposes a bidirectional Transformer with a cloze task predicting the masked target items for SR.

**2). Semantic-based Baselines: UniSRec** (Hou et al. [2022]) utilizes the associated description text of items to learn transferable representations across different recommendation scenarios, using an enhanced mixture-of-experts adaptor to enhance domain fusion and adaptation. **VQ-Rec** (Hou et al. [2023]) maps item text to a vector of discrete indices for learning transferable sequential recommenders. **TIGER** (Rajput et al. [2024]) trains a Transformer-based sequence-to-sequence model with semantic IDs obtained from RQ-VAE to enhance its generalization ability.

**3). Generative Baselines: ACVAE** (Xie et al. [2021]) proposes an adversarial and contrastive variational autoencoder for SR combining the ideas of CVAE and GAN. **DiffuRec** (Li et al. [2023b])

**Table 2:** Performance of different models. Bold (underline) is used to denote the best (second-best) metric, and '*' indicates significant improvements relative to the best baseline (t-test P<.05). 'R@K' ('N@K') is short for 'Recall@K' ('NDCG@K'). The features of items have been listed, whether ID, text (T), or both (ID+T).

| Methods | Scientific | | | | Office | | | | Online Retail | | | |
|---------|------|------|------|------|------|------|------|------|------|------|------|------|
| | R@10 | N@10 | R@50 | N@50 | R@10 | N@10 | R@50 | N@50 | R@10 | N@10 | R@50 | N@50 |
| SASRec$_T$ | 0.1049 | 0.0527 | 0.1754 | 0.0683 | 0.1047 | 0.0714 | 0.1638 | 0.0857 | 0.1461 | 0.0674 | 0.3781 | 0.1186 |
| BERT4Rec$_{ID}$ | 0.0473 | 0.0258 | 0.1092 | 0.0394 | 0.0798 | 0.0605 | 0.1207 | 0.0717 | 0.1354 | 0.0661 | 0.3517 | 0.1159 |
| UniSRec$_T$ | 0.1104 | 0.0537 | 0.1890 | 0.0787 | 0.1024 | 0.0621 | 0.1668 | 0.0798 | 0.1274 | 0.0598 | 0.3294 | 0.1054 |
| VQ-Rec$_T$ | 0.1129 | 0.0577 | 0.2046 | 0.0749 | 0.1090 | 0.0676 | 0.1714 | 0.0845 | 0.1532 | 0.0713 | 0.3975 | 0.1254 |
| TIGER$_T$ | 0.1057 | 0.0597 | 0.1803 | 0.0682 | 0.1056 | 0.0712 | 0.1597 | 0.0868 | 0.0745 | 0.0390 | 0.2216 | 0.0701 |
| ACVAE$_{ID}$ | 0.0463 | 0.0315 | 0.0906 | 0.0457 | 0.0549 | 0.0397 | 0.1003 | 0.0519 | 0.0884 | 0.0410 | 0.1897 | 0.0648 |
| DiffuRec$_{ID}$ | 0.1145 | 0.0594 | 0.1907 | 0.0752 | 0.1056 | 0.0689 | 0.1781 | 0.0832 | 0.0402 | 0.0189 | 0.0849 | 0.0321 |
| DreamRec$_{ID+T}$ | 0.0845 | 0.0421 | 0.1645 | 0.0688 | 0.0954 | 0.0557 | 0.1662 | 0.0694 | 0.0577 | 0.0261 | 0.0997 | 0.0544 |
| **DDSR$_T$** | **0.1207*** | **0.0663*** | **0.2153*** | **0.0842*** | **0.1138*** | **0.0768*** | **0.1926*** | **0.0925*** | **0.1687*** | **0.0876*** | **0.4021** | **0.1322*** |
| Improv. | +5.41% | +11.61% | +5.23% | +6.99% | +4.40% | +8.14% | +6.46% | +7.93% | +10.12% | +22.86% | +1.16% | +5.42% |

introduces the diffusion model into the field of SR reconstructing target item representation from a Transformer backbone with the user's historical interaction behaviors. **DreamRec** (Yang et al. [2024]) uses the historical interaction sequence as conditional guiding information for the diffusion model to enable personalized recommendations.

**Evaluation Settings**. Following previous works Hou et al. [2022], Zhao et al. [2022], Zhou et al. [2020], we evaluate all models using metrics Recall@K and NDCG@K, and report experimental results for $K = 10, 50$. We employ the leave-one-out strategy for performance evaluation across all datasets. Concretely, we consider the last interaction as the test set, the second-to-last interaction as the validation set, and all preceding interactions as the training set. The ground-truth item of each sequence is ranked among all the other items while evaluating (Krichene and Rendle [2022]). The implementation details of DDSR are illustrated in Appendix C.1.

## 5.2 Overall Performance

In this section, we compare the performance of DDSR with baseline models in terms of Top-$K$ recommendation accuracy under consistent experimental conditions (same data preprocessing), as summarized in Table 2. For models that recommend based on item IDs, we provide semantic information to them by jointly utilizing fixed text embeddings obtained from pre-trained BERT and the embeddings corresponding to item IDs in the model's embedding layer, to ensure fairness in the experimental setup. For all models, the final table records the better of the three methods, using only ID, only text, or both text and ID.

We observe that text-enhanced SR methods (UniSRec, VQ-Rec, TIGER) tend to benefit from textual information, leading to improved performance compared to conventional methods in most cases. Notably, VQ-Rec, employing discrete semantic encoding, generally outperforms UniSRec, which relies on continuous text embeddings, across various settings. This is despite UniSRec already using techniques like parameter whitening and MoE-enhanced Adaptor to enhance textual information. We posit that an excessive emphasis on text similarity can yield suboptimal outcomes, while the conversion to codes mitigates the coupling between items and semantic information. The corresponding representations of the codes are relearned in the sequence-to-sequence model, allowing them to include more sequential structural information. While similarly based on discrete semantic encoding, the performance of TIGER is not stable. We do not rule out the possibility that there may be discrepancies between our implementation and the actual model, as it has not been open-sourced. Furthermore, we attribute the instability to TIGER's semantic ID length, which is limited to only 4 characters, potentially insufficient for expressing complex information.

In methods grounded in generative models, the performance of DiffuRec and DreamRec, based on diffusion models, surpasses that of ACVAE, relying on GAN and VAE. This disparity arises from the inherent advantages of diffusion models over VAE and GAN, as they circumvent the issue of posterior collapse, wherein the generated hidden representations lack crucial information about user preferences. Notably, DiffuRec achieves superior performance despite its limited capacity to handle semantic information, yet it still exhibits recommendation performance comparable to VQ-Rec. This suggests that diffusion models can yield effective hidden representations of items and users.

**Table 3:** Ablation analysis of DDSR. Bold font indicates the best metric.

| Variants | Scientific | | | | Office | | | | Online Retail | | | |
|---|---|---|---|---|---|---|---|---|---|---|---|---|
| | R@10 | N@10 | R@50 | N@50 | R@10 | N@10 | R@50 | N@50 | R@10 | N@10 | R@50 | N@50 |
| Uniform transition | **0.1207** | **0.0663** | **0.2153** | **0.0842** | 0.1097 | 0.0752 | 0.1889 | **0.0931** | 0.1517 | 0.0724 | 0.3967 | 0.1273 |
| Importance transition | 0.1192 | 0.0631 | 0.2110 | 0.0813 | **0.1138** | **0.0768** | **0.1926** | 0.0925 | **0.1687** | **0.0876** | **0.4021** | **0.1322** |
| w/o diffusion | 0.1126 | 0.0563 | 0.2059 | 0.0749 | 0.1076 | 0.0659 | 0.1729 | 0.0851 | 0.1549 | 0.0719 | 0.4003 | 0.1268 |
| RQ-VAE ID | 0.1195 | 0.0645 | 0.1987 | 0.0825 | 0.1017 | 0.0609 | 0.1611 | 0.0784 | 0.1081 | 0.0564 | 0.2772 | 0.0853 |
| Random ID | 0.0554 | 0.0307 | 0.1193 | 0.0487 | 0.0548 | 0.0386 | 0.0982 | 0.0501 | 0.0428 | 0.0220 | 0.0863 | 0.0334 |

In all three datasets, DDSR achieves significant improvements, demonstrating the effectiveness of our approach. We attribute this success to two key factors. Firstly, the integration of semantic information mitigates data sparsity issues, as evidenced by the enhanced performance of semantic-based models on smaller datasets compared to traditional recommendation methods. Secondly, training on fuzzy sets generated by discrete diffusion furnishes the model with additional information. This is consistent with our theoretical analysis in Section 3.3, which posits that the diffused information space constitutes a completion of the original space, rendering models built on this enriched space effectively solvable. Moreover, while DiffuRec, relying on a continuous state space diffusion model, exhibits instability when confronted with larger and more intricate datasets, DDSR maintains a distinct advantage. We attribute this to DDSR's retention of the discrete space without transitioning it into a continuous one and introducing noise, thus circumventing the loss of meaningful information.

## 5.3 Ablation Study

We analyze the impact of semantic ID and discrete diffusion on final performance and conduct an ablation study to compare the results under different settings, as shown in Table 3. The Uniform transition and importance transition are two discrete diffusion methods provided in Section 3.2, corresponding to different transition matrices. To control variables, these methods, along with the non-diffusion case, are all applied using semantic ID obtained through PQ. It can be observed that on larger datasets, the importance transition has relatively more advantages, and both methods outperform the non-diffusion scenario.

To control variables and accurately evaluate the performance of IDs, diffusion is applied in the last two rows of the experiments (the method based on PQ ID with diffusion corresponds to the first and second rows, so it is not listed again). The PQ ID and RQ-VAE ID in the table corresponds to the two methods of obtaining semantic identifiers provided in Section 4.1. Random ID represents using a randomly generated codebook to replace the semantic identifiers, where the random identifier of item $c_i$ is simply $c_i = (c_{i,1}, \ldots, c_{i,m})$ with $c_{i,j}$ uniformly sampled from $1, 2, ..., K$. Using Random ID means the model no longer gains semantic information. As observed, PQ ID exhibits greater stability and outperforms RQ-VAE ID across multiple datasets. Nevertheless, RQ-VAE ID requires less memory space due to its ability to achieve satisfactory representation with fewer codebook lengths. Semantic identifiers consistently outperform the Random ID, underscoring the significance of leveraging content-based semantic information. Indeed, models utilizing Random ID even underperform compared to SASRec. This can be attributed to SASRec's approach of setting independent embeddings for each item, rather than blending unrelated embeddings.

## 5.4 Further Analysis

**Performance Analysis on Cold-Start Items**. In this study, we evaluate the efficacy of DDSR in recommending cold-start items. Generating effective item embeddings without item information poses a challenge for SR models. To evaluate this, we partition the test data into two groups based on item popularity. For the Office dataset, the range [0, 5) demarcates long-tail Items, whereas for the Online Retail dataset, it is [0, 20). All other items are classified as Popular Items. The results are presented in Figure 2a. Notably, DDSR and VQ-Rec demonstrate substantial improvement over SASRec, which solely leverages a Transformer, particularly for long-tail Items, also referred to as the cold-start group. This is attributed to the integration of semantic information, enabling the model to acquire prior knowledge about items to some extent. Furthermore, DDSR demonstrates even greater enhancement compared to VQ-Rec in cold-start scenarios. We attribute this to the discrete diffusion method, which introduces a 'fuzziness' effect in the interaction records, facilitating the inclusion of items with fewer interactions in the training process.

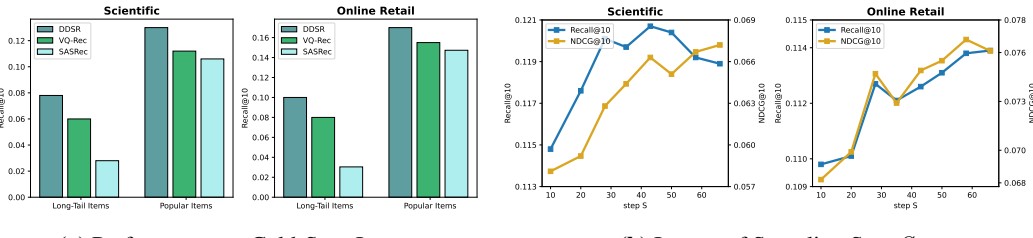

**(a)** Performance on Cold-Start Items.  **(b)** Impact of Sampling Step $S$.

**Figure 2:** Impact of Item Popularity and Sampling Step $S$.

**Impact of Sampling Step** $S$. The sampling step $S$ represents the diffusion step number divided by the number of inference steps executed simultaneously, rounded down to the nearest integer. We illustrate in Figure 2b the influence of various sampling step settings. The model demonstrates optimal performance with approximately 50 sampling steps, with a slight increase for more complex datasets, albeit without significant disparities. Excessive sampling steps prolong the inference time without commensurate performance improvements, while inadequate steps lead to decreased performance.

**Efficiency Analysis.** We compared the time complexity and specific running overhead of DDSR with several other baseline algorithms, and the detailed results can be found in Appendix C.2.

## 6    Discussion

We proposed the DDSR model for the sequential recommendation, which employed discrete diffusion to construct fuzzy sets of user interaction sequences. This process was iteratively refined during inference, utilizing the sampling formula for discrete diffusion to derive the ultimate recommendation outcomes. Notably, although DDSR had borrowed the form of diffusion and sampling over time steps from diffusion models, it fundamentally differed from directly using diffusion models. If we viewed sequential recommendation through the lens of causality, the interaction sequence was the 'cause' and the recommended item was the 'effect'. Diffusion models typically address the target, blurring the 'effect', whereas DDSR has blurred the 'cause', inspired by the theory of fuzzy information processing. Although dual assurances, both theoretical and experimental results, have been provided to substantiate the superior performance of DDSR, it is imperative to recognize its inherent limitations. Despite our efforts to implement efficient computational methods, the nature of diffusion and sampling processes inevitably results in reduced efficiency and increased time complexity. Potential refinements, such as approximating the diffusion process and accelerating the sampling algorithm, could offer effective strategies, which we will explore in future work.

## 7    Acknowledgments

This work is supported by the National Natural Science Foundation of China (No. U23A20319, 62202443). Hao Wang also thanks the CCF-Tencent Rhino-Bird Open Research Fund (RAGR20230124).

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

# A    Proof of the theory

**Theorem A.1.** *The model often has the following form:*

$$f : (U \times V, \rho) \to (U' \times V', \rho),$$

*which requires the model to be a continuous mapping from the original input-output space (which can also be the original space). The data distribution on the aggregated model set can be modeled:*

$$f' : (D, \rho') \to (U' \times V', \rho),$$

*defined as $\gamma$ such that $f' = \gamma(\mu)$. When the static distribution approximates the dynamic distribution, $\mu$ is the extended static scattering coefficient; otherwise, it is the linear information distribution coefficient $\gamma$, ensuring that the determination of $f'$ is unique. Therefore, $(D, \rho')$ is the completion space of $(U \times V, \rho)$, consisting of the complete set of $(U \times V, \rho)$ and its separation.*

*In summary, the information diffusion space must be the completion space of the original space. Due to the same dimensionality as the original space, the modeling on the aggregated set is reliable.*

*Next, consider the model (prediction model or simulation model) built on $D(X)$. In practical applications, the input-output set is generally constructed in two ways based on the original data:*

$$e_i = \frac{x_i - x_{\min}}{x_{\max} - x_{\min}},$$

*where $e_i$ is the normalized sequence; $x_i$ is the original sequence data; $x_{\min}$ is the minimum value of the original sequence; $x_{\max}$ is the maximum value of the original sequence. It is obvious that $e_i \in [0, 1]$, meaning $(U \times V) \subseteq [0, 1]$, and the function boundary on $D(X)$. When evaluating risk, the original sequence data generates input-output sets, which clearly have bounded intervals $x_i \in [x_{\min}, x_{\max}]$, meaning the function boundary on $D(X)$ is also bounded. Therefore, $D(X)$ is necessarily a subset of the real number set.*

*In the prediction model, the attribute function obtained from the aggregated set is:*

$$\mu_{y_0} = \sum_u \mu_{x_0}(u)\mu_{A_1}(u, v), \quad u \in U, v \in V. \tag{6}$$

*Among them, $\mu_{R_i}(u) = \mu_{A_i}(u)\mu_{A_i}(v), \quad u \in U, v \in V$. Since the output sequence and the input sequence are generated by the same $\{e_i\}$, and $\mu_i = x_{i+1}$:*

*Separately substituting into (6), we can obtain:*

$$\mu_{x_0} \cdot g(x_0) = \sum_u \mu_{x_0}\mu_{A_i}\mu_{B_i} \tag{7}$$

*or*

$$\mu_{x_0} + g(x_0) = \sum_u \mu_{x_0}\mu_{A_i}\mu_{B_i}.$$

*Rearranging, we get:*

$$\mu_{x_0} = \sum_u \frac{\mu_{x_0}\mu_{A_i}\mu_{B_i}}{g(x_0)} = \sum_u G(x_0, \mu_{x_0}),$$

*and*

$$\mu_{x_0} = \sum_u \mu_{x_0}\mu_{A_i}\mu_{B_i} - g(x_0) = \sum_u \left[ \mu_{x_0}\mu_{A_i}\mu_{B_i} - \frac{g(x_0)}{n} \right] = \sum_u G(x_0, \mu_{x_0}). \tag{8}$$

*The initial value problem for the differential equation $\dot{y}(t) = f(t, y(t)), y(t_0) = y_0$ can be transformed into an integral equation:*

$$y(t) = \int_u f(t, y(t))dt + y_0, \quad t \in u; \tag{9}$$

*Since the cumulative sum and substitution integrals can be used in the scatter system, (8) is the integral equation's discrete model on $D(X)$.*

$$|G(x_i) - G(x_j)| = |G(x_i, \mu_{x_i}) - G(x_j, \mu_{x_j})| = \begin{cases} |\mu_{x_i} \cdot g(x_i) - \mu_{x_j} \cdot g(x_j)|, & X \subseteq N(0,1) \\ |\mu_{x_i} + g(x_i) - \mu_{x_j} + g(x_j)| \end{cases}$$

(10)

*Both cases satisfy:*

$$K = \max_{x \in U} \frac{\Delta}{\mu_{x_i} - \mu_{x_j}} \text{ such that } |G(x_i) - G(x_j)| \leq K |x_i - x_j|,$$

*i.e., the function $G(X)$ on $D(X)$ satisfies the Lipschitz condition, thus the attribute function must have a solution.*

*Since the attribute function has a solution on $D(X)$, it can be deduced by the "maximum" principle that the predicted output value must be:*

$$\bar{y_0} = \left( \sum_{i=1}^{n} w_i v_i' \right) / \left( \sum_{i=1}^{n} w_i \right),$$

*where the weights $w_i = \mu_{y_0}(v') = \max_{v \in V} \{\mu_{y_0}(v)\}$.*

*Therefore, the information diffusion approximation model is established.*

# B  Model Supplement

We first obtain fixed text embeddings from the relevant descriptions of items (e.g., product descriptions, item titles, or brands) using the pre-trained BERT model. Specifically, for an item $v_i$ with a corresponding description $\{w_1, w_2, \ldots, w_c\}$, the corresponding embedding vector is $e_i = \text{BERT}([[CLS]; w_1, \ldots, w_c]) \in \mathbb{R}^{d_W}$, where "[; ]" denotes the concatenation operation. Next, obtaining Semantic IDs from $e_i$ can be achieved through the following two methods:

**Product Quantization (PQ).** Similar to the VQ-Rec approach, we evenly divided $e_i$ into $m$ sub-vectors $e_i = [e_{i,1}; \ldots; e_{i,m}]$, each with a dimension of $d_W/m$. Denote $a_{p,j} \in \mathbb{R}^{d_W/m}$ as the $j$-th centroid embedding in the codebook corresponding to the $p$-th sub-vector. For each sub-vector, the index of the nearest centroid from the corresponding PQ codebook is selected to form its discrete code $c_{i,p} = \arg\min_j \|e_{i,p} - \boldsymbol{a}_{p,j}\|^2 \in \{1, 2, \ldots, K\}$. The centroids are obtained through the commonly used Optimized Product Quantization (OPQ) method. Finally, $c_i = (c_{i,1}, \ldots, c_{i,m})$ is used as the semantic ID for item $v_i$.

**RQ-VAE.** RQ-VAE generate a set of codewords by quantizing the residuals. First, the input $e_i$ is encoded into $r_0$ using encoder **E**. Next, similar to PQ, the nearest centroid to it in the first codebook, assumed to be $a_{p,1}$, is found and its index is taken to form the first discrete code $c_{i,1}$. The residual is defined as $r_1 := r_0 - a_{p_1,1}$. The same operation is performed to obtain $c_{i,2}$ and this process is repeated $m$ times to obtain the complete Semantic ID. RQ-VAE use loss function $\mathcal{L}(\boldsymbol{x}) := \mathcal{L}_{\text{recon}} + \mathcal{L}_{\text{rqvae}}$ jointly trains the encoder, decoder, and the codebook, where $\mathcal{L}_{\text{recon}} := \|e_i - \widehat{e}_i\|^2$ is reconstruction loss, $\mathcal{L}_{\text{rqvae}} := \sum_{d=0}^{m-1} \|\text{sg}[\boldsymbol{r}_i] - a_{p_i,i}\|^2 + \beta\|\boldsymbol{r}_i - \text{sg}[a_{p_i,i}]\|^2$. Here $\widehat{e}_i$ is the output of the decoder, and sg is the stop-gradient operation Van Den Oord and Vinyals [2017]. Because the norm of the residuals decreases progressively, the importance of the codebooks obtained in this manner also diminishes with each level. Alternatively, it can be said that the encoding at each position has varying levels of granularity. In our experiments, we found that this approach requires a smaller codebook size than PQ, but it is slightly less stable.

In fact, once the codebook is established, it remains fixed throughout the subsequent model training process. Therefore, the quality of the codebook directly affects the training outcomes. With the rapid development of large language models, considering the use of more advanced pre-training schemes to replace BERT could be an effective way to enhance performance. However, this is not the focus of our current research, we leave this topic for future exploration.

**Table 4:** Time complexity analysis of various models, 'Comp.' is an abbreviation for 'Complexity'.

| Model | SASRec | BERT4Rec | UniSRec | TIGER | DiffuRec | DDSR |
|---|---|---|---|---|---|---|
| **Comp.** | $O(nd^2 + dn^2)$ | $O(nd^2 + dn^2)$ | $O(nd^2 + dn^2)$ | $O(mnd^2 + mdn^2)$ | $O(nd^2 + dn^2)$ | $O(mnd^2 + mdn^2)$ |

## C  Experimental Supplement

### C.1  Implementation Details

All of our experiments were conducted on a single RTX 4090. We implemented our models based on PyTorch and the popular open-source recommendation library RecBole. We used $(m = 32) \times (K = 256)$ as the code representation scheme for PQ IDs and $(m = 6) \times (K = 256)$ for RQ-VAE IDs. For baseline models, to ensure fair comparison, we optimized all methods using the Adam optimizer and searched for hyperparameters to find the best results. Since we did not find open-source code for the TIGER model, we attempted to replicate it as faithfully as possible based on its paper; however, its performance may have been slightly lower in some experiments due to difficulties in unifying some details. For UniSRec and VQ-Rec, we utilize their code in a form that does not involve pretraining with the entire dataset, as we aim to evaluate all baselines and DDSR considering recommendations on a single dataset rather than cross-domain. The batch size was set to 2,048. The learning rate was adjusted among $\{0.0001, 0.0002, 0.0003, 0.0005, 0.001\}$. The model achieving the highest NDCG@10 result on the validation set was selected for evaluation on the test set. We employed an early stopping strategy with a patience of 10 epochs.

Regarding diffusion, all models were trained on a diffusion process of 1000 steps, and the time step embeddings are implemented using cosine embeddings, similar to the work of Li et al. [2023b]. For the diffusion with uniform transition, we employ the cosine schedule proposed by Hoogeboom et al. [2021] to set the transition probabilities $(1 - \beta_t)$. For the diffusion with importance sampling, we adopt a linear schedule similar to the one used in Ho et al. [2020], where $\sigma^2$ increases linearly from $10^{-4} * K$ to $0.02 * K$. Skip steps in the sampling process were chosen among $\{100, 50, 35, 28, 23, 20, 17, 15\}$. While there have been many works on accelerating sampling in continuous state space diffusion models, the development in discrete diffusion is still insufficient. Here, we adopted a basic uniform skip scheme for more efficient and effective sampling, which is one of our next research directions. If the evaluation steps do not divide 1000 evenly, the last step may be skipped.

### C.2  Efficiency Analysis

We list the time complexities of six baselines in Table 4, where $n$ denotes sequence length, $d$ is the dimension of the hidden layers, and $m$ is the length of the codebook used when semantic IDs are utilized. Most of these models are based on the transformer or its variants, hence the time complexity is $O(nd^2 + dn^2)$. Only TIGER and our proposed DDSR model are trained using semantic IDs, making their time complexity $m$ times that of other methods. We plan to make further improvements in our subsequent work.

In addition, we compared the actual operational costs of the DDSR model with those of UniSRec and DiffuRec, as shown in Table 5. Although we adopted the method of 'performing inference for $k$ steps at a time', which reduced the sampling steps and lowered the evaluation time compared to DiffuRec, we must acknowledge that DDSR still has certain limitations in terms of operational costs, necessitating further improvements in future work.

## D  Diffusion Models in Continuous State Space

### D.1  DDPM

Diffusion models comprise a forward diffusion process and a backward denoising process. We begin with the widely recognized denoising diffusion probabilistic model (DDPM) Ho et al. [2020]. We start by defining our data distribution $x_0 \sim q(x_0)$ and a Markovian noising process $q$ which gradually adds noise to the data $x_0$ to produce noised samples $x_T$. In particular, each step of the noising process

**Table 5:** Comparison of Actual Operational Costs.

| Datasets | Model | GPU memory (GB) | Training Time (s/epoch) | Evaluation Time (s/epoch) |
|---|---|---|---|---|
| Scientific | UniSRec | 8.32 | 3.51 | 0.67 |
| | DiffuRec | 14.94 | 4.97 | 17.52 |
| | DDSR | 12.41 | 6.76 | 11.38 |
| Office | UniSRec | 8.29 | 9.96 | 1.13 |
| | DiffuRec | 14.85 | 25.81 | 127.41 |
| | DDSR | 12.48 | 36.19 | 69.10 |
| Online Retail | UniSRec | 9.96 | 52.19 | 3.70 |
| | DiffuRec | 15.97 | 65.22 | 103.44 |
| | DDSR | 13.47 | 83.51 | 60.11 |

adds Gaussian noise according to a variance schedule given by $\beta_t$:

$$q(x_t \mid x_{t-1}) := \mathcal{N}(x_t; \sqrt{1 - \beta_t} x_{t-1}, \beta_t \mathbf{I})$$

Furthermore, $q(x_t \mid x_0)$ can be expressed as a Gaussian distribution. With $\alpha_t := 1 - \beta_t$ and $\bar{\alpha}_t := \prod_{s=0}^{t} \alpha_s$, $q(x_t \mid x_0) = \mathcal{N}(x_t; \sqrt{\bar{\alpha}_t} x_0, (1 - \bar{\alpha}_t)\mathbf{I}) = \sqrt{\bar{\alpha}_t} x_0 + \sqrt{1 - \bar{\alpha}_t}\epsilon$, where $\epsilon \sim \mathcal{N}(0, \mathbf{I})$. Here, $1 - \bar{\alpha}_t$ indicates the variance of the noise at an arbitrary timestep, and this can be used to define the noise schedule instead of $\beta_t$.

Using Bayes theorem, one finds that the posterior $q(x_{t-1}|x_t, x_0)$ is also a Gaussian with mean $\tilde{\mu}_t(x_t, x_0)$ and variance $\tilde{\beta}_t$ defined as follows:

$$\tilde{\mu}_t(x_t, x_0) := \frac{\sqrt{\alpha_{t-1}}\beta_t}{1 - \alpha_t} x_0 + \frac{\sqrt{\alpha_t}(1 - \alpha_{t-1})}{1 - \alpha_t} x_t \tag{8}$$

$$\tilde{\beta}_t := \frac{1 - \alpha_{t-1}}{1 - \alpha_t} \beta_t \tag{9}$$

$$q(x_{t-1}|x_t, x_0) = \mathcal{N}(x_{t-1}; \tilde{\mu}_t(x_t, x_0), \tilde{\beta}_t \mathbf{I}) \tag{10}$$

If we wish to sample from the data distribution $q(x_0)$, we can first sample from $q(x_T)$ and then sample reverse steps $q(x_{t-1}|x_t)$ until we reach $x_0$. Under reasonable settings for $\beta_t$ and $T$, the distribution $q(x_T)$ is nearly an isotropic Gaussian distribution, so sampling $x_T$ is trivial. All that is left is to approximate $q(x_{t-1}|x_t)$ using a neural network, since it cannot be computed exactly when the data distribution is unknown. To this end, Sohl-Dickstein et al. [56] note that $q(x_{t-1}|x_t)$ approaches a diagonal Gaussian distribution as $T \to \infty$ and correspondingly $\beta_t \to 0$, so it is sufficient to train a neural network to predict a mean $\mu_\theta$ and a diagonal covariance matrix $\Sigma_\theta$:

$$p_\theta(x_{t-1}|x_t) := \mathcal{N}(x_{t-1}; \mu_\theta(x_t, t), \Sigma_\theta(x_t, t)) \tag{11}$$

To train this model such that $p_\theta(x_0)$ learns the true data distribution $q(x_0)$, we can optimize the following variational lower-bound $L_{\text{vlb}}$ for $p_\theta(x_0)$:

$$L_{\text{vlb}} := L_0 + L_1 + \ldots + L_{T-1} + L_T \tag{12}$$

$$L_0 := -\log p_\theta(x_0|x_1) \tag{13}$$

$$L_{t-1} := D_{KL}(q(x_{t-1}|x_t, x_0)\|p_\theta(x_{t-1}|x_t)) \tag{14}$$

$$L_T := D_{KL}(q(x_T|x_0)\|p(x_T)) \tag{15}$$

While the above objective is well-justified, Ho et al. [25] found that a different objective produces better samples in practice. In particular, they do not directly parameterize $\mu_\theta(x_t, t)$ as a neural network, but instead train a model $\epsilon_\theta(x_t, t)$ to predict $\epsilon$ from Equation 17. This simplified objective is defined as follows:

$$L_{\text{simple}} := \mathbb{E}_{t \sim [1,T], x_0 \sim q(x_0), \epsilon \sim \mathcal{N}(0,\mathbf{I})} \left[ \| \epsilon - \epsilon_\theta(x_t, t) \|_2^2 \right] \tag{16}$$

During sampling, we can use substitution to derive $\mu_\theta(x_t)$ from $\epsilon_\theta(x_t, t)$:

$$\mu_\theta(x_t) = \frac{1}{\sqrt{\alpha_t}} \left( x_t - \frac{1 - \alpha_t}{\sqrt{1 - \alpha_t}} \epsilon_\theta(x_t, t) \right) \tag{17}$$

Note that $L_{\text{simple}}$ does not provide any learning signal for $\Sigma_\theta(x_t, t)$. Ho et al. [25] find that instead of learning $\Sigma_\theta(x_t, t)$, they can fix it to a constant, choosing either $\beta_t$ or $\tilde{\beta}_t$. These values correspond to learning noise and the reverse process variance respectively.

## D.2 Score-Based Generative Model

In this section, we introduce a Score-Based Generative Model (SGMs) Song et al. [2020], specifically a diffusion model represented in the form of Stochastic Differential Equations (SDEs). SGMs model the forward diffusion process using the stochastic differential equation:

$$dx_t = f(x_t, t)dt + g(t)dw, \boldsymbol{x}_0 \sim p_0 = p_{\text{target}}, \tag{18}$$

where $t \in [0, T]$, and $w$ signifies Brownian motion, $p_{\text{target}}$ represents target distribution. The function $f(\cdot, t) : \mathbb{R}^d \to \mathbb{R}^d$ is a vector-valued function called the drift coefficient of $x(t)$, and $g(\cdot) : \mathbb{R} \to \mathbb{R}$ is a scalar function known as the diffusion coefficient of $x(t)$. The functions $f$ and $g$ determine the type of prior distribution $p_{\text{prior}}$ to which the forward process will diffuse, and they are typically designed to make the prior distribution a Gaussian distribution. As a remarkable result from Anderson [1982], the reverse of the diffusion process is also a diffusion process, given by the following reverse-time SDE:

$$dx_t = [f(x_t, t) - g(t)^2 \nabla_x \log p_t(x)]dt + g(t)d\bar{w}, \tag{19}$$
$$\boldsymbol{x}_T \sim p_T \approx p_{\text{prior}},$$

where $\bar{w}$ is a standard Wiener process in reverse time. The term $\nabla_x \log p_t(x)$, which represents the score function of the marginal density $p_t$, is the only unknown term in this reverse process. SGMs learns its approximate target $s_\theta(x(t), t)$ through denoising score matching (DSM) Hyvärinen and Dayan [2005], with $s_\theta$ referred to as the denoising model:

$$\theta^* = \arg\min_\theta \mathbb{E}_{t \sim U(0,T)} \mathbb{E}_{x(0)} \mathbb{E}_{x(t)|x(0)}$$
$$\left[ \| s_\theta(x(t), t) - \nabla_x \log p_{0t}(x|x(0)) \|^2 \right]. \tag{20}$$

Here, $\lambda(t)$ is a positive weighting coefficient, $t \sim \mathcal{U}(0, T)$. The joint distribution $p_{0t}(x|x(0))$ is the conditional transition distribution from $x(0)$ to $x(t)$, which is determined by the pre-defined forward SDE. To summarize, SGMs first utilize the diffusion process defined in Equation(18) to obtain the distribution $x_t$ at intermediate time steps. Then, they minimize the loss defined in Equation(20) to train the denoising model $s_\theta$ and sample iteratively using the formula defined in Equation(19) to obtain the final result.

