# OpenReview forum: "Breaking Determinism: Fuzzy Modeling of Sequential Recommendation Using Discrete State Space Diffusion Model"
_NeurIPS.cc/2024/Conference — NeurIPS 2024 poster_

### Official Review · Reviewer_LQPH · 2024-07-12

**Soundness:** 4
**Presentation:** 3
**Contribution:** 3
**Rating:** 7
**Confidence:** 4

**Summary:**

The paper presents the DDSR model for Sequential Recommendation (SR), which better captures user interest evolution by using fuzzy sets of interaction sequences. Unlike traditional methods, DDSR effectively handles the unpredictability of user behavior and addresses cold start issues. Experiments on benchmark datasets show that DDSR outperforms existing methods, demonstrating its effectiveness in SR tasks.

**Strengths:**

1. DDSR introduces a novel perspective in SR, effectively addressing the limitations of traditional sequential modeling methods and enhancing recommendation accuracy.
2. DDSR uses fuzzy sets of interaction sequences and diffusion transition processes in discrete state spaces, improving the model's ability to capture the randomness and unpredictability of user behavior.
3. The theoretical justification for constructing the information diffusion approximation model is sound and fundamental.
4. Quantitative experimental results demonstrate the model's effectiveness, with particularly exciting results in cold start scenarios.

**Weaknesses:**

1. The paper lacks an algorithm to describe the entire model operation. Including this would greatly enhance readers' understanding of the model and its theoretical underpinnings.
2. All experimental results are quantitative. It is recommended to supplement with a case study or visual experiment.
3. Although the author has analyzed the time complexity, I believe most readers would also appreciate a comparison of actual running times.

**Questions:**

See comments above.

**Limitations:**

This paper has discussed limitations.

---

> ### Author Rebuttal · Authors · 2024-08-05
>
> First of all, thank you very much for reading our work carefully and for your valuable comments and suggestions, from which we have greatly benefited. Next, I will explain and respond to the shortcomings you have pointed out, and we will make corresponding revisions in the official version of our work based on your reminders.
>
> W1: Regarding a clearer description of our algorithm, we think your suggestion is very reasonable. To avoid making the theoretical part too lengthy within the main discussion, we placed the description of discrete diffusion separately in Section 3. However, this approach may not facilitate readers' understanding of the overall workflow. We add our algorithm flowchart below and will incorporate it into the official version of our paper based on your suggestion.
>
> **Training of DDSR.**
>
> **Input:**
> - Historical Interaction Sequence of user $u$:  $v_{1:n-1} = c_{1:n-1;1:m}$ ;
> - Target item:  $v_n = c_{n;1:m}$;
> - Transition matricest: $Q_t$;
> - Approximator: $f_{\theta}(\cdot)$.
>
> **Output:**
> - Well-trained Approximator: $f_{\theta}(\cdot)$.
>
> **Procedure:**
> 1. **while** not converged **do**
> 2. &emsp; Sample Diffusion Time:  $t \sim [0,1,...,T]$;
> 3. &emsp; Calculate $t$-step transition probability: $\quad\overline{\boldsymbol{Q}}_t=\boldsymbol{Q}_1\boldsymbol{Q}_2\ldots\boldsymbol{Q}_t$;
> 4. &emsp; Convert $c_{n;1:m}$ to one-hot encoding $x_{n;1:m}^0$;
> 5. &emsp; Obtain the discrete state $x_{n;1:m}^t$ after $t$ steps by Equation (2),  thereby obtain the 'fuzzy set' $c_{1:n-1;1:m}^{t}$;
> 6. &emsp; Modeling $c_{2:n;1:m}$ based on 'fuzzy sets' through Equation (5);
> 7. &emsp; Take gradient descent step on $\nabla$ $L_{CE}$ ($\hat{c}_{2:n;1:m}$, $c_{2:n;1:m}$).
>
>
> **Sampling of DDSR.**
>
> **Input:**
>    - Historical Sequence: $v_{1:n-1} = c_{1:n-1;1:m}$
>    - Well-trained Approximator: $f_{\theta}(\cdot)$
>    - Sampling Steps: $T$.
>
> **Output:**
>    - Predicted Target Item: $v_{n}$
>
> 1. Let $x_T$ = $c_{1:n-1;1:m}$;
>
> 2. Let t = T;
>
> 3. **while** $t>0$ **do**
>
> 4. &emsp;  Use the trained $f_{\theta}(\cdot)$ to obtain predictions $\widetilde{x}_{0}$ with $x_t$ and $t$  as inputs;
>
> 5. &emsp;  Substitute $\widetilde{x}_{0}$ into equation (7) to obtain the distribution of $t-1$ step;
>
> 6.  **end while**
>
> 7. $\widetilde{v}_{n}$ =
> $x_0$[-1;1:m];
>
> 8. if   the same code project exists: $v_n$ = $\widetilde{v}_{n}$;
>
> &emsp; else: $v_n$ is the project in the space closest to $\widetilde{v}_{n}$.
>
>
> W2: Regarding the issues with the experimental section, we greatly appreciate your constructive suggestions. We recognize that our experimental setup was not comprehensive, so we have added a study on the impact of codebook length on recommendation performance. We will present the following experimental results on dataset Scientific in the form of bar charts in the paper:
> | **code_len** | **Recall@10** | **NDCG@10** | **Recall@50** | **NDCG@50** | **GPU memory (GB)** |**Training Time (s/epoch)** |**Evaluation Time (s/epoch)** |
> |----------------|---------------------|-----------------------------|-------------------------------|-------------------------------|-------------------------------|-------------------------------|-------------------------------|
> | **64** |0.1235|0.0656|0.2396|0.0907|22.14|13.56|19.40|
> | **32** | 0.1207|0.0663|0.2153 |0.0842 |12.41|6.76|11.38|
> | **16** |0.1145|0.0603|0.2161|0.0846|9.74|4.34|3.79|
> | **8** |0.1084|0.0589 |0.2184| 0.0829|8.37|2.55|3.61|
> | **4** |0.0836|0.0433 |0.1820| 0.0759|7.43|2.49|3.33|
>
> W3: Regarding the comparison of actual runtime, we have compared the GPU memory usage and runtime of the UniSRec, DiffuRec, and our DDSR model on a single A100 GPU. The specific results are as follows:
> | **Datasets**   | **Model** | **GPU memory (GB)** | **Training Time (s/epoch)** | **Evaluation Time (s/epoch)** |
> |----------------|-----------|---------------------|-----------------------------|-------------------------------|
> | **Scientific**  | UniSRec   | 8.32                | 3.51                        | 0.67|
> |                | DiffuRec  | 14.94               | 4.97                       | 17.52|
> |                | DDSR    | 12.41               | 6.76                        | 11.38|
> | **Office**   | UniSRec   | 8.29                | 9.96                        | 1.13|
> |                | DiffuRec  | 14.85               | 25.81                       | 127.41 |
> |                | DDSR    | 12.48              | 36.19                       | 69.10   |
> | **Online Retail**| UniSRec   | 9.96                | 52.19                      | 3.70  |
> |                | DiffuRec  | 15.97               | 65.22                       | 103.44                        |
> |                | DDSR    | 13.47               | 83.51                       | 60.11                          |
> The increased runtime of DDSR primarily stems from the introduction of the diffusion model and the increased time complexity due to the conversion of project embeddings into discrete codes. The current time complexity is $O(mnd^2+mdn^2)$, which is $m$ (codebook length) times that of methods like SASRec that only use the Transformer model. Fortunately, we have found in our experiments that good results can be achieved with relatively smaller settings for dimension $d$, as we now effectively use a vector of length $m*d$ to represent a project. Additionally, the use of discrete codes helps reduce storage overhead and DDSR enables faster sampling speeds compared to other diffusion model-based methods by performing inference for k steps at a time. We believe that the initial efficiency issues are still within an acceptable range. Additionally, since state transitions within the discrete space do not introduce noise, we found in our experiments that DDSR's training has significantly enhanced stability and convergence speed compared to DiffuRec. For example, DiffuRec requires about one to two hundred epochs to converge on the Scientific and Office datasets, while our model only needs about forty to fifty epochs.

---

> > ### Comment · Reviewer_LQPH · 2024-08-14
> >
> > Thanks for the response which havs addessed my main concerns, and thus I keep my original score of acceptance.

---

> > > ### Author Response · Authors · 2024-08-14
> > >
> > > Thank you for your positive feedback and for taking the time to review our work. We greatly appreciate your thoughtful comments and are glad that we could address your concerns. Your support and acceptance are highly valued.
> > > Best regards！

---

### Official Review · Reviewer_ukuQ · 2024-07-15

**Soundness:** 2
**Presentation:** 3
**Contribution:** 3
**Rating:** 5
**Confidence:** 4

**Summary:**

This paper studies the problem of fuzzy modeling for sequential recommendation. The work proposes to leverage the fuzzy sets of interaction sequences for modeling the nature of users' evolving real interests. It also introduces the discrete diffusion modeling specifically for the discrete data. The experiment results showcase the better model performance in comparison with several included baselines.

**Strengths:**

1. This paper studies the optimization of sequential recommender systems, which is an important application for the information retrieval domain.
2. The paper is written with fair clarity and thus it is easy to follow.
3. The paper provides careful analyses to support the claims made by this paper and the experimental results generally demonstrate the effectiveness of the proposed methods.

**Weaknesses:**

1. Insufficient baseline comparison for the conventional sequential recommender models, as the two included models are outdated.
2. Insufficient empirical analysis in terms of the model efficiency. Firstly, the runtime cost comparison is missing, which is however very important to recommender system prototype evaluation for practical deployment consideration. Secondly, based on the complexity analysis, it is crucial to showcase the performance variations (effectiveness/efficiency) by varying the codebook length $m$.

**Questions:**

1. How many times of performance evaluations are conducted for the average results?
2. What are the real runtime costs for the proposed model and other baselines for comparison?

**Limitations:**

Please see the weakness for details.

---

> ### Author Rebuttal · Authors · 2024-08-06
>
> Firstly, we sincerely appreciate your careful review of our work and the extremely valuable suggestions you have made! In response to the issues you have pointed out, we will improve our research and provide detailed answers to each of your questions, hoping to clearly address your concerns.
>
> W1: Regarding the concern you raised about the insufficient baseline comparison with traditional sequence recommendation models, we indeed acknowledge that there is a gap in our work in this area.  During our research, we primarily focused on comparing with other text-based and generative model-based sequential recommendation approaches, which may have led to an inadequate evaluation of traditional models.  To address this, we have added experiments with the following three traditional sequential recommendation models:
>
> 1. **HSTU**[1]: A novel sequence recommendation architecture introduced by Meta in May this year. HSTU utilizes a hierarchical sequence transduction framework, incorporating a modified attention mechanism and a micro-batching algorithm (M-FALCON) for efficient handling of high-cardinality and dynamic recommendation data.
>
> 2. **Mamba4Rec**[2]: A model proposed by Liu et al. in June this year, based on the latest Mamba design, which includes a series of sequence modeling techniques.
>
> 3. **FDSA**[3]: The feature-level deep self-attention network introduced by Zhang et al., which integrates various heterogeneous features of items into a feature sequence with different weights through a fundamental mechanism.
>
> Here are our experimental results:
> | **Datasets**   | **Model** | **Recall@10** | **NDCG@10** | **Recall@50** | **NDCG@50** |
> |----------------|-----------|---------------------|-----------------------------|-------------------------------|-------------------------------|
> | **Scientific**  | HSTU| 0.1086        |0.0543| 0.1816| 0.0721|
> || Mamba4Rec |0.0976  |  0.0562  |  0.1881 |0.0760   |
> || FDSA    |0.0901| 0.0599 |  0.1683|  0.0766 |
> | **Office**   | HSTU   | 0.1102   | 0.0718|0.1709| 0.0872  |
> | | Mamba4Rec  |0.1079|0.0727  |    0.1633   |     0.0858 |
> || FDSA    | 0.1108|0.0730|0.1723|0.0854 |
> | **Online Retail**| HSTU   |0.1456| 0.0698|0.3750  | 0.1213  |
> || Mamba4Rec  | 0.1410              |  0.0686                  | 0.3783                   |   0.1188                           |
> || FDSA   |           0.1478   |          0.0719            |              0.3817  |                0.1229             |
>
> Among these, we have integrated HSTU into our framework, which based on RecBole, as its original source code did not include a test set partition. Thank you for your suggestion; we will enhance the baseline comparisons in the formal version of the paper.
>
> [1] Zhai J, Liao L, Liu X, et al. Actions speak louder than words: Trillion-parameter sequential transducers for generative recommendations[J]. arXiv preprint arXiv:2402.17152, 2024.
>
> [2] Liu C, Lin J, Wang J, et al. Mamba4rec: Towards efficient sequential recommendation with selective state space models[J]. arXiv preprint arXiv:2403.03900, 2024.
>
> [3] Zhang T, Zhao P, Liu Y, et al. Feature-level deeper self-attention network for sequential recommendation[C]//IJCAI. 2019: 4320-4326.
>
> Q1&W2: Thank you for pointing that out. We will address these omissions in the final version of the paper. Below are the actual runtime cost comparison results of our DDSR model with UniSRec and DiffuRec:
> | **Datasets**   | **Model** | **GPU memory (GB)** | **Training Time (s/epoch)** | **Evaluation Time (s/epoch)** |
> |----------------|-----------|---------------------|-----------------------------|-------------------------------|
> | **Scientific**  | UniSRec   | 8.32  | 3.51  | 0.67|
> | | DiffuRec  | 14.94| 4.97 | 17.52 |
> | | DDSR    | 12.41| 6.76 | 11.38 |
> | **Office**   | UniSRec   | 8.29 | 9.96 | 1.13 |
> |  | DiffuRec  | 14.85 | 25.81| 127.41|
> | | DDSR    | 12.48  | 36.19| 69.10 |
> | **Online Retail**| UniSRec   | 9.96| 52.19| 3.70|
> | | DiffuRec  | 15.97| 65.22 | 103.44|
> | | DDSR    | 13.47 | 83.51| 60.11 |
>
> Regarding the performance variations with different codebook lengths that you mentioned, we have added experiments here. Taking the Scientific dataset as an example, we explore how codebooks obtained through quantization methods affect the recommendation performance at various lengths：
> | **code_len** | **Recall@10** | **NDCG@10** | **Recall@50** | **NDCG@50** | **GPU memory (GB)** |**Training Time (s/epoch)** |**Evaluation Time (s/epoch)** |
> |----------------|---------------------|-----------------------------|-------------------------------|-------------------------------|-------------------------------|-------------------------------|-------------------------------|
> | **64** |0.1235|0.0656|0.2396|0.0907|22.14|13.56|19.40|
> | **32** | 0.1207|0.0663|0.2153 |0.0842 |12.41|6.76|11.38|
> | **16** |0.1145|0.0603|0.2161|0.0846|9.74|4.34|3.79|
> | **8** |0.1084|0.0589 |0.2184| 0.0829|8.37|2.55|3.61|
> | **4** |0.0836|0.0433 |0.1820| 0.0759|7.43|2.49|3.33|
>
> For the Scientific dataset alone, a codebook length of 64 achieved best results most of the time. We regret not having made sufficient attempts before.
>
> Q1: During our experiments, we ran each model five times and took the average as the final result to validate the significance of improvements. We failed to mention this point in the implementation details section of the paper, which was an oversight. We are very sorry for this and will include this information in the final version of the paper.
>
> If you believe that our response has resolved the issues raised and alleviated your concerns, we kindly request that you consider adjusting the score accordingly. If you still have any doubts or suggestions, we earnestly ask you to point them out to further enhance our work. Once again, thank you for your thoughtful comments and consideration.

---

> ### Author Response · Authors · 2024-08-12
>
> Dear Reviewer,
>
> I hope this message finds you well. I am writing to inquire about the progress of my rebuttal submission. I understand that the review process can be quite involved, but I would appreciate any updates you could provide regarding the status of the review.
>
> Thank you very much for your time and effort!
>
> Best regards！

---

> > ### Author Response · Authors · 2024-08-14
> >
> > Dear Reviewer,
> >
> > I hope this message finds you well. I understand that you may have a busy schedule, and I truly appreciate the time and effort you are dedicating to reviewing our manuscript.
> >
> > As the deadline for feedback approaches, I wanted to kindly check in and see if there is any additional information or clarification we can provide to assist with the review process. Your insights are invaluable to us, and we look forward to any feedback you might have.
> >
> > Thank you again for your time and consideration.
> >
> > Best regards！

---

> > > ### Comment · Reviewer_ukuQ · 2024-08-14
> > >
> > > Dear Authors,
> > >
> > > Thanks for the detailed responses. They have addressed my major concerns, and I will increase my score to positive rating. Please make sure the additional experiments will be included in the revision for publication.

---

> > > > ### Author Response · Authors · 2024-08-14
> > > >
> > > > Dear Reviewer,
> > > >
> > > > Thank you for your recognition of our detailed responses and for increasing the rating. We will ensure that the additional experiments are included in the revised manuscript and reflected in our submission. We sincerely appreciate your valuable feedback, support, and the time and effort you have devoted to our work once again！
> > > >
> > > > Best regards！

---

### Official Review · Reviewer_6uVb · 2024-07-15

**Soundness:** 4
**Presentation:** 3
**Contribution:** 3
**Rating:** 8
**Confidence:** 4

**Summary:**

The paper presents a diffusion model-based sequential recommendation from a novel information-theoretical perspective, which operates on discrete structural state spaces along with semantic labels improving efficiency and tackling cold-start issues.

**Strengths:**

Strengths*
1.	Novelty. The paper uses a directed graph to model sequential recommendation and models the transitions of nodes with discrete diffusion. Besides, authors introduce semantic tags to replace meaningless item IDs, enhancing efficiency and improving cold start issues.
2.	Soundness. The paper presents enough theoretic analysis and experimental evaluations to validate the proposed method. The significant performance improvement on three public datasets, along with a series of further ablation study and analysis, clearly demonstrates the soundness of the paper.
3.	Easy-followed. The paper presents sufficient algorithmic detail. In addition, the authors provide sufficient codes to help understand and re-product the algorithm.
Weaknesses*
1.	Citation mixed in the text make obstacles for reading. See questions for detail.
2.	Writing should be more regulated. See questions for detail.

**Weaknesses:**

See Above

**Questions:**

1.	In text such as line 235, 236, 251, the citation text is literally repeated.
2.	In text such as line 20-26, it would be better to add brackets to distinguish references from the main text.
3.	In line 74 and 144, it should be explicit that “2” and “4.1” refers to “Section 2” and “Section 4.1”.
4.	The usage and format of quotation marks is a bit confusing, see line 34, 219 and Table 1 caption for examples.

**Limitations:**

There is no significant limitation. Perhaps the authors could consider give a rough discussion or comment on how the computational complexity (or the coefficient constant) of the proposed method can be improved.

---

> ### Author Rebuttal · Authors · 2024-08-06
>
> We sincerely appreciate the valuable time you have dedicated to our work and the encouragement you have given us! Thank you for thoroughly reading our paper and pointing out some issues, which indeed arose due to our oversight; for this, we deeply apologize. In the final version, we will carefully review and correct each issue you have raised and perform a comprehensive proofreading from start to finish to ensure that such problems do not recur.
>
> Regarding the limitations you've mentioned, specifically the need to discuss improvements in computational complexity, we indeed recognize the importance of this issue and plan to address it in the final version of our paper. We believe that there are two feasible approaches for improvement: through model architecture and coding strategies. For instance, within the model architecture, one could employ linear attention mechanisms in Transformers as a substitute, redesigning the computation of $K$ and $V$ matrices to reduce the time complexity of the attention mechanism from $O(n^2d)$ to $O(nd^2)$. Additionally, we have taken note of the recently proposed mamba model, which maintains linear complexity while demonstrating robust performance. From a coding perspective, exploring ways to convey sufficient information with shorter codebook lengths seems to be an effective strategy. For example, in our research, the codebook lengths generated by RQ-VAE are significantly shorter than those required by traditional quantization methods, yet maintaining effectiveness on this basis remains a critical challenge.
>
> Lastly, we greatly appreciate the valuable suggestions you have provided.
>
> [1]Gu A, Dao T. Mamba: Linear-time sequence modeling with selective state spaces[J]. arXiv preprint arXiv:2312.00752, 2023.
>
> [2]Angelos Katharopoulos, Apoorv Vyas, Nikolaos Pappas, and François Fleuret. “Transformers are RNNs: Fast Autoregressive Transformers with Linear Attention”. In: International Conference on Machine Learning. PMLR. 2020,
> pp. 5156–5165.

---

> > ### Comment · Reviewer_6uVb · 2024-08-13
> > **To authors**
> >
> > You have addressed some of the concerns I raised in my previous review. I believe this work is indeed novel and meets the standards required for this conference. As a result, I am inclined to raise my score.
> > Best regards,

---

> > > ### Author Response · Authors · 2024-08-13
> > >
> > > Dear Reviewer,
> > >
> > > We would like to once again express our sincere gratitude for the time and effort you invested in reviewing our manuscript. We deeply appreciate your constructive feedback and are thankful for the higher evaluation during the rebuttal phase. Your insights have not only enhanced our work but also encouraged us to further refine our research. Based on your suggestions, we have made appropriate revisions to our paper accordingly.
> > >
> > > Thank you once again for your thoughtful review and for recognizing the potential of our work. We are extremely grateful for your expert guidance and assistance.
> > >
> > > Best regards!

---

### Official Review · Reviewer_qbmo · 2024-07-17

**Soundness:** 2
**Presentation:** 3
**Contribution:** 2
**Rating:** 5
**Confidence:** 3

**Summary:**

The paper presents a new model for sequential recommendation (SR) called DDSR, which aims to predict items of interest for users based on their past behavior. The authors critique conventional SR methods for not adequately capturing the randomness and unpredictability in user behavior. DDSR uses fuzzy sets of interaction sequences to better reflect the true evolution of user interests. Unlike common diffusion models that operate in continuous domains, DDSR uses diffusion transition processes in discrete state spaces, avoiding information loss by employing structured transitions. To tackle the inefficiency of matrix transformations in large discrete spaces, the model utilizes semantic labels derived from quantization or RQ-VAE instead of item IDs. Tests on three public datasets show that DDSR surpasses current state-of-the-art methods, proving its effectiveness in SR tasks.

**Strengths:**

Effective combination of pre-existing ideas: discrete diffusion with semantic IDs. Empirical results in Section 5 on Scientific, Office, Online Retail datasets.

**Weaknesses:**

I think a main weakness of the paper is a lack of clarity in writing. The method is still unclear to me, although this may be partly due to my lack of experience in recommender systems. I would condense and combine Sections 3 and 4 and move Section 2 to end. The technical novelty is also not entirely clear to me. Is it in the form of the transition matrices used for the discrete diffusion? Or is the technical novelty claimed that the items are recommended based on the output of running a few steps of discrete diffusion (adding noise to the features)?

**Questions:**

Could the authors please elaborate on the analogy between fuzzy sets and discrete diffusion?

**Limitations:**

Yes, limitations adequately addressed.

---

> ### Author Rebuttal · Authors · 2024-08-05
>
> Firstly, we appreciate the valuable time you have invested in our work and the constructive suggestions you have offered; we will rigorously revise our paper based on your feedback. We hope the following explanations will somewhat alleviate your concerns.
>
> W1: We find your suggestion to relocate Section 2 to the end of the paper quite insightful, and we plan to adopt this adjustment in the final version. Regarding the idea of combining Sections 3 and 4, we need to explain that the original intention of discussing Section 3 before Section 4 separately was to separate the theoretical and model sections. This was aimed at preventing the model section from becoming too lengthy, which could negatively affect the reader's experience. However, your advice has highlighted a potential compromise in clarity that we had not fully appreciated. To enhance our discussion, we will introduce an algorithm flowchart below as an initial step. In the formal version, we will thoughtfully consider how to integrate your feedback. Currently, we are contemplating including Sections 3.1 and 3.2 within the methods part of Chapter 4 and positioning Section 3.3 just before the experimental segment, to provide theoretical backing for our proposed DDSR model. We would appreciate your thoughts on whether this revised structure seems appropriate. Here is the algorithm flowchart we provided (after generating discrete semantic encoding).
>
> **Training of DDSR.**
>
> **Input:**
> - Historical Interaction Sequence of user $u$:  $v_{1:n-1} = c_{1:n-1;1:m}$ ;
> - Target item:  $v_n = c_{n;1:m}$;
> - Transition matricest: $Q_t$;
> - Approximator: $f_{\theta}(\cdot)$.
>
> **Output:**
> - Well-trained Approximator: $f_{\theta}(\cdot)$.
>
> **Procedure:**
> 1. **while** not converged **do**
> 2. &emsp; Sample Diffusion Time:  $t \sim [0,1,...,T]$;
> 3. &emsp; Calculate $t$-step transition probability: $\quad\overline{\boldsymbol{Q}}_t=\boldsymbol{Q}_1\boldsymbol{Q}_2\ldots\boldsymbol{Q}_t$;
> 4. &emsp; Convert $c_{n;1:m}$ to one-hot encoding $x_{n;1:m}^0$;
> 5. &emsp; Obtain the discrete state $x_{n;1:m}^t$ after $t$ steps by Equation (2),  thereby obtain the 'fuzzy set' $c_{1:n-1;1:m}^{t}$;
> 6. &emsp; Modeling $c_{2:n;1:m}$ based on 'fuzzy sets' through Equation (5);
> 7. &emsp; Take gradient descent step on $\nabla$ $L_{CE}$ ($\hat{c}_{2:n;1:m}$, $c_{2:n;1:m}$).
>
>
> **Sampling of DDSR.**
>
> **Input:**
>    - Historical Sequence: $v_{1:n-1} = c_{1:n-1;1:m}$
>    - Well-trained Approximator: $f_{\theta}(\cdot)$
>    - Sampling Steps: $T$.
>
> **Output:**
>    - Predicted Target Item: $v_{n}$
>
> 1. Let $x_T$ = $c_{1:n-1;1:m}$;
>
> 2. Let t = T;
>
> 3. **while** $t>0$ **do**
>
> 4. &emsp;  Use the trained $f_{\theta}(\cdot)$ to obtain predictions $\widetilde{x}_{0}$ with $x_t$ and $t$  as inputs;
>
> 5. &emsp;  Substitute $\widetilde{x}_{0}$ into equation (7) to obtain the distribution of $t-1$ step;
>
> 6.  **end while**
>
> 7. $\widetilde{v}_{n}$ =
> $x_0$[-1;1:m];
>
> 8. if   the same code project exists: $v_n$ = $\widetilde{v}_{n}$;
>
> &emsp; else: $v_n$ is the project in the space closest to $\widetilde{v}_{n}$.
>
>
> W2: I apologize for the lack of clarity in our description and hope that the following explanation will address your concerns about the technical innovations. Unlike traditional diffusion models that add noise to features, we use discrete diffusion to induce structured transformations among discrete nodes, such as semantic labels, within a discrete state space using transition matrices. This approach replaces the introduction of noise in conventional diffusion models and avoids the deviation of representations towards meaningless directions. The semantic space itself is inherently meaningful, and the discrete diffusion framework allows us to design controllable transition schemes, such as importance-based transitions, which are a major source of performance improvement. We also note that traditional diffusion models add noise to embeddings, which often exacerbates training difficulties in highly sparse scenarios like sequence recommendation. For instance, in our experiments, we observed that DiffuRec required over three hundred training epochs to converge on most datasets. Our work is also the first attempt to enhance diffusion models with textual information in the context of recommendation.
>
>
> Q1: We introduce the concept of fuzzy sets because diffusion models are inherently designed for generative tasks, which fundamentally differ in form from autoregressive tasks like sequence recommendation, where the $n$th item is predicted using the previous $n-1$ items. By employing fuzzy modeling to redefine the diffusion process, we ensure that the diffusion can be theoretically applied to the target embeddings but also enhances adaptability without compromising the robust theoretical support inherent to diffusion models. As for discrete diffusion, it represents a form of fuzzy modeling in our work. The specific reasons for using discrete diffusion are presented in W1.
>
> Thank you for your thorough review. If you feel our clarifications have indeed alleviated any doubts, we kindly ask you to consider adjusting the score accordingly. Should you have any further questions or require additional clarification, please do not hesitate to point them out.

---

> > ### Comment · Reviewer_qbmo · 2024-08-13
> > **Response to Rebuttal**
> >
> > I thank the authors for their thoughtful modifications to the paper - is it possible to upload this revision to OpenReview before the review period ends? I think the inclusion of the pseudo-code for training and sampling algorithms is a big step forward. Also, with the inclusion of the new experimental results I am glad to increase my score.

---

> > > ### Author Response · Authors · 2024-08-14
> > >
> > > Dear Reviewer,
> > >
> > > Thank you very much for your positive feedback and for recognizing the improvements in our revised manuscript. We greatly appreciate your support, particularly regarding the inclusion of the pseudo-code and the new experimental results.
> > >
> > > We will promptly upload the revised version to OpenReview before the review period ends, as per your suggestion.
> > >
> > > Thank you again for your valuable guidance and for increasing your score.
> > >
> > > Best regards！

---

> ### Author Response · Authors · 2024-08-12
>
> Dear Reviewer,
>
> I hope this message finds you well. I am writing to inquire about the progress of my rebuttal submission. I understand that the review process can be quite involved, but I would appreciate any updates you could provide regarding the status of the review.
>
> Thank you very much for your time and effort!
>
> Best regards！

---

### Decision · Program_Chairs · 2024-09-25

**Decision:**

Accept (poster)

**Comment:**

The paper introduces an information theory-based improvement of sequential modeling to capture the randomness in user behavior. The paper was well-received by all reviewers and inspired (what looks like) a fruitful exchange between the reviewers and the authors who addressed the reviewer concerns. Most reviewers increased their scores after the rebuttal. To meet the expectations of the reviewers, the authors should incorporate the rebuttal in their final paper, including improving the clarity of the paper, discussion of computational complexity, and adding more experimental results.